# Heteroepitaxy Growth and Characterization of High-Quality AlN Films for Far-Ultraviolet Photodetection

**DOI:** 10.3390/nano12234169

**Published:** 2022-11-24

**Authors:** Titao Li, Yaoping Lu, Zuxin Chen

**Affiliations:** 1Jinjiang Joint Institute of Microelectronics, College of Physics and Information Engineering, Fuzhou University, Fuzhou 350108, China; 2School of Semiconductor Science and Technology, South China Normal University, Foshan 528225, China

**Keywords:** AlN, heteroepitaxy, far-ultraviolet photodetection, single crystalline films, MOCVD

## Abstract

The ultra-wide bandgap (~6.2 eV), thermal stability and radiation tolerance of AlN make it an ideal choice for preparation of high-performance far-ultraviolet photodetectors (FUV PDs). However, the challenge of epitaxial crack-free AlN single-crystalline films (SCFs) on GaN templates with low defect density has limited its practical applications in vertical devices. Here, a novel preparation strategy of high-quality AlN films was proposed via the metal organic chemical vapor deposition (MOCVD) technique. Cross-sectional transmission electron microscopy (TEM) studies clearly indicate that sharp, crack-free AlN films in single-crystal configurations were achieved. We also constructed a p-graphene/i-AlN/n-GaN photovoltaic FUV PD with excellent spectral selectivity for the FUV/UV-C rejection ratio of >10^3^, a sharp cutoff edge at 206 nm and a high responsivity of 25 mA/W. This work provides an important reference for device design of AlN materials for high-performance FUV PDs.

## 1. Introduction

Far-ultraviolet (100–200 nm) photodetectors (FUV PDs) based on ultra-wide-bandgap semiconductors (UWBSs) have shown potential in the fields of space science, radiation monitoring, biomedicine and the electronics industry [1,2,3,4,5]. AlN is an ideal material for the preparation of FUV PDs, owing to its ultra-wide bandgap (~6.2 eV), thermal stability and radiation resistance [6,7,8]. At present, AlN-based FUV PDs have two kinds of device structure: horizontal or vertical, depending on the direction of carrier transmission between the two electrodes. The horizontal structure has the advantage of easy preparation, whereas the vertical structure has the characteristics of zero power consumption, ultra-fast response speed and high sensitivity, owing to its built-in electric field, which can satisfy the increasing demand of high-performance FUV photodetection [9]. Among the many conductive substrates, the commercial high-quality GaN template is a suitable choice for preparation of vertical structures as the charge transport layers, as it has the same hexagonal symmetry structure as AlN and a relatively low lattice mismatch of 2.4% [10]. However, few studies on AlN heteroepitaxy on GaN substrates have been reported because the practical application requirements are rarely covered, and such heteroepitaxy represents a challenge.

Because the lattice mismatch between sapphire and GaN can be as high as 17%, directly epitaxial GaN and AlN films on sapphire inevitably leads to high surface roughness and poor crystal quality of AlN and GaN [11]. An effective way to reduce the lattice mismatch is to insert an AlN transition layer (TL) between GaN and sapphire [12]. Even if a high-quality GaN template has been obtained, the heteroepitaxial AlN single-crystalline films (SCFs) on GaN could crack the AlN and decompose the GaN, owing to the reduced lattice constant (a_AlN_ = 0.3112 nm) and increased optimum epitaxial temperature (T_AlN_ = 1250 ℃) of AlN relative to that of GaN (a_GaN_ = 0.3189 nm, T_GaN_ = 1050 ℃) [13] as a result of low photoresponsivity and high dark current of AlN-based devices. According to analysis of the growth characteristics of AlN, the key to effectively avoiding the above two risks is to modulate the growth temperature and the thickness of the top AlN epilayer. An excessive a growth temperature (1050 ℃ < T < 1250 ℃) usually leads to the decomposition of GaN, whereas an excessively low growth temperature (T < 1050 ℃) will not endow AlN atoms with sufficient energy for transverse diffusion, resulting in failure of the growth mode transition from three-dimensional (3D) to two-dimensional (2D). Excessive thickness of an AlN epilayer on GaN usually leads to cracks.

This work is devoted to proposing a novel preparation strategy for crack-free, low-defect-density AlN SCFs on high-quality GaN to achieve high-performance AlN-based FUV PDs. A crack-free AlN SCF with low defect density was successfully obtained by metal organic chemical vapor deposition (MOCVD) technique by inserting an AlN TL between GaN and sapphire, followed by an isothermally epitaxial 200 nm AlN. On the basis of AlN SCF, a photovoltaic FUV PD was fabricated by adopting graphene as the transparent conductive layer of the device, which was discovered in our previous work [14]. As expected, the device exhibits low dark current, high responsivity of 25 mA/W and a FUV/UV-C rejection ratio of >10^3^ at 0 V bias, which is significantly higher than that of previously reported FUV PDs. The high detection performance of the device shows application potential in space exploration. Our material preparation concept and device design scheme also provide a basic reference for researchers interested in FUV PDs.

## 2. Experimental Section

### 2.1. Preparation of AlN Films

Using an Aixtron MOCVD system with radio frequency (RF) heating, AlN films were grown on a 2-inch diameter c-plane sapphire substrate. During the growth process, trimethyl aluminum, trimethyl gallium and ammonia were used as the sources of aluminum, gallium and nitrogen, respectively. High-purity hydrogen (H_2_) was used as carrier gas. First, a 20 nm thick AlN buffer layer was deposited at 900 ℃; then, a 300 nm AlN layer (V/III molar ratio of 500) was grown at a high temperature of 1250 ℃. Subsequently, GaN with a thickness of 1200 nm was grown at 1050 ℃. Finally, 200 nm AlN was grown by pulsed atomic-layer epitaxy (PALE) at 1050 °C to avoid GaN layer decomposition.

### 2.2. Device Preparation

First, 5 × 5 mm^2^ graphene was transferred onto an AlN surface by wet transfer method. Then, 20 nm thick circular Ti electrodes (300 μm diameter) and 50 nm thick Au electrodes were deposited successively using thermal evaporation technology. Finally, indium was hot-melted on the n-GaN side as the negative electrode.

### 2.3. Characterizations

HRTEM, XRD (HRXRD) spectra, AFM and UV transmittance were obtained by an FEI Talos F200 S, Bruker D8 Advance X-ray diffractometer (Bruker, Billerica, MA, USA), Brooker Dimension Edge atomic force microscope (Bruker, Billerica, MA, USA) and UV-visible-near infrared (UV-VIS-NIR) spectrophotometer (Shimadzu UV-3600 Plus, Shimadzu, Kyoto, Japan), respectively. The PL spectra were collected at room temperature using 193 nm Exacted ArF EX5/250 mini excimer laser (GAM LASER, Orlando, FL, USA) with a power density of 0.64 W/cm^2^ (50 Hz); spectral acquisition was dependent on a QE65PRO Scientific-Grade spectrometer (Ocean Optics, Dunedin, FL, USA) with H70 grating (200–289 nm).

### 2.4. Response Test of Devices

The I-V and I-*t* curve were measured by a Keithley 4200A-SCS source meter (Keithley, Cleveland, OH, USA), with a 185 nm monochromatic light provided by the spectral line of a quartz-encapsulated low-pressure mercury lamp. For the FUV spectral response test, we adopted a deep UV spectral response test system, including an IHR320 (Horiba, Kyoto, Japan) deep UV monochromator, an L11798 deuterium lamp light source (Hamamatsu, Hamamatsu, Japan) and a Keithley 6517B (Keithley, Cleveland, OH, USA) source meter to measure the photocurrent.

## 3. Results and Discussion

Large lattice mismatch usually leads to a high density of threading dislocations (TDs), which usually act as non-radiative recombination centers affecting the internal quantum efficiency (IQE) in the device [12,15]. Therefore, AlN single-crystalline films with low defect density are required for the preparation of high-performance FUV PDs. The classical heteroepitaxial “two-step method” has achieved considerable success in the heteroepitaxy of nitride SCFs and commercial blue LED devices [12]. Herein, an MOCVD two-step growth method is adopted. Firstly, a buffer layer (nucleation layer) is regulated to effectively release the stress. In this step, the appropriate growth temperature and thickness of the nucleation layer have an important influence on the quality of the subsequent epitaxial crystallization layer. Secondly, the epitaxial conditions are regulated at high temperature to controlling the timing of the growth mode transition from 3D to 2D; thus, the crystalline grain is effectively healed, and most of the misfit dislocations and stacking faults are “filtered out” [16,17]. Finally, the density of the defect is considerably reduced as a result of a successful synthesis of a high-quality SCF. Figure 1a shows a brightfield cross-sectional transmission electron microscopic (TEM) image of the heterostructure with a 20 nm AlN buffer, 300 nm AlN TL, a 1.2 μm Si-doped GaN template and a 200 nm upper AlN layer. According to the cross-sectional TEM results, no obvious cracks occurred in the upper AlN layer. Generally, XRD analysis is used to investigate crystallinity [18,19]. The crystal orientation analysis of AlN and GaN via XRD pattern (Figure 1b) exhibits clear characteristic diffraction peaks. In the photoluminescence (PL) spectrum shown in Figure 1c, the dominated band-edge emission peak centered at ~209 nm (with a narrow full width at half maximum (FWHM) of 5.2 nm) shows a high intensity ratio relative to that of the defect peaks (at longer wavelengths), indicating that the rate of nonradiative recombination in the structures is not sufficiently high to completely suppress radiative recombination. Figure 1d shows an atomic-force microscopic (AFM) image of the AlN SCF with typical step-flow morphology and a root mean square roughness surface of 0.21 nm, indicating an atomically smooth surface and high crystalline quality in the AlN. The crack-free and smooth characteristics can be also confirmed by the three-dimensional AFM topography, as shown in Figure 1e. The dislocation density of AlN film can be further calculated by the X-ray rocking curve. Figure 1f shows the rocking curve of (0002) plane of the AlN top layer. The 166 arcsec FWHM of the (0002) plane is lower than that of currently reported state-of-the-art AlN film (on GaN, 220 arcsec), indicating that the screw dislocation density was modulated a very low level. According to the value of FWHM, the screw-type dislocation density is calculated to be 6.0 × 10^7^ cm^−2^ by the formula N_screw_ = FWHM^2^/(4.35 × b^2^) [20], where b = 〈0001〉 is the Burgers vector of a screw dislocation in AlN. The low defect density of AlN SCF helps to construct an FUV PD with high detection performance [21,22].

High-resolution transmission electron microscopy (HRTEM) can be used to study the microstructure of the heterojunction interface [23,24,25,26]. Figure 2a shows cross-sectional HRTEM images of the sample recorded at the adjacent AlN/GaN interface. There is no obvious misfit dislocation at the interface (a fast Fourier transformation image from the masked area of Figure 2a is shown in Figure 2b), indicating that the lattice has not completely relaxed. The details of the crystalline quality and orientation of the AlN/GaN heterojunction interface were characterized by selected-area electron diffraction (SAED), as shown in Figure 2 c. The interface is composed of two sets of slightly staggered clear and bright diffraction spots, indicating that AlN and GaN have the same crystal orientation and high crystal quality. Figure 2d is a cross-sectional HRTEM image of the top AlN layer away from the interface at high magnification. Clear and regularly arranged atoms prove the high crystalline quality of AlN film. Figure 2e shows a SAED image of the AlN layer near the surface. The bright and regular circular spots also prove the high crystalline quality of the top AlN SCF, which is consistent with the results shown in Figure 1.

Because the built-in electric field can quickly separate photogenerated carriers, photovoltaic FUV PDs with a typical p-i-n structure have the advantages of faster response speed, higher sensitivity and zero power consumption relative to photoconductive FUV PDs [27,28]. Owing to its high mobility and high UV transmittance, p-graphene is an ideal charge transport layer to achieve high open-circuit voltage and ultrafast photoresponse of photovoltaic FUV PDs [14,29]. Based on the above heterostructure, a p-type graphene was transferred onto the AlN film by wet method to construct a photovoltaic PD with a co-directional double-heterojunction ‘p-i-n’ structure (Figure 3a). P-graphene acts as the upper transparent conductive window layer of the PD, and the photosensitive AlN SCF layer is sandwiched between the upper conductive p-graphene layer and the lower conductive n-GaN layer. Figure 3b depicts the typical I-V characteristic curve of the device in the dark state at room temperature. The nanoampere-level current indicates that crack-free AlN contributes to the reduction in dark current. In addition, the standard rectification characteristics shown in the curve are the addition of p-Graphene/AlN and AlN/n-GaN homogeneous heterojunctions, as shown in the inset of Figure 3b.

Figure 4a depicts the current–voltage (I-V) characteristics of the device under 185 nm steady-state FUV light at a power of 30.6 μW/cm^2^ and dark conditions. The results show that the device has an open-circuit voltage (V_OC_) of ~1.15 V and an ultra-high switching ratio at different voltages. High V_OC_ is one of the most important indicators of the high optoelectrical conversion efficiency of devices [30,31]. High V_OC_ is mainly attributed to the enhancement effect of quasi-Fermi-level splitting and the high crystalline quality of AlN film [29].

The primary indicator of FUV PDs is the ability to respond to FUV signals exclusively as a result of excellent wavelength selectivity. We evaluated the FUV spectral selectivity of the device at 0 V bias, as shown in Figure 4b. The wavelength-dependent photoresponsivity (R_λ_) curve is based on continuously tunable monochromatic light from deep UV to FUV. R_λ_ represents the detection capability of the device, defined as R_λ_ = ΔI/PS, where ΔI is the photocurrent minus the dark current, P is the incident light power density and S is the light absorption area. The peak responsivity (25 mA/W) of the device occurs at 194 nm, with a high external quantum efficiency (EQE) of 16%. Additionally, the R_λ_-λ curve of the device shows a very sharp cutoff edge at approximately 206 nm, and the FUV (194 nm)/UV-C (212 nm) rejection ratio is as high as 1.95 × 10^3^ (Figure 4b inset), which confirms the excellent spectral selectivity of the FUV PD. Such a sharp cutoff edge is the result of the high crystalline quality of AlN SCF [3]. The relevant response performance of the device is better than that of most of currently reported FUV PDs, as shown in Table 1 [32,33,34,35,36].

The switching characteristics of the FUV PD were further evaluated by I-*t* test, as shown in Figure 4c,d. The device has very sensitive switching characteristics to the incident FUV light at varying optical powers, and the photocurrent increases significantly with increased optical power density (Figure 4c). The photocurrent of the device is also related to the applied bias. Figure 4d shows the I-t curves at 1 V and 2 V bias, respectively. As the bias voltage increases, the photocurrent continues to increase, mainly because the high external bias voltage can increase the drift speed of the photogenerated electron–hole pair and suppress the recombination probability of the photogenerated electron–hole pair.

## 4. Conclusions

In summary, herein, we proposed a novel preparation strategy for crack-free, low-defect-density AlN SCFs on a GaN template by MOCVD epitaxial technique. AlN SCFs with an atomically flat surface were successfully prepared, showing high crystal quality. A semiconductor FUV PD was fabricated based on an AlN/GaN heterojunction combined with graphene as a transparent conductive window. The device exhibits excellent wavelength selectivity for the detection of FUV light and shows an ultra-high FUV (194 nm)/UV-C (212 nm) rejection ratio of >10^3^. Meanwhile, the high open-circuit voltage of 1.15 V, good switching characteristics and high responsivity of 25 mA/W all show that the device has broad application prospects for highly sensitive FUV photodetection.

## Figures and Tables

**Figure 1 nanomaterials-12-04169-f001:**
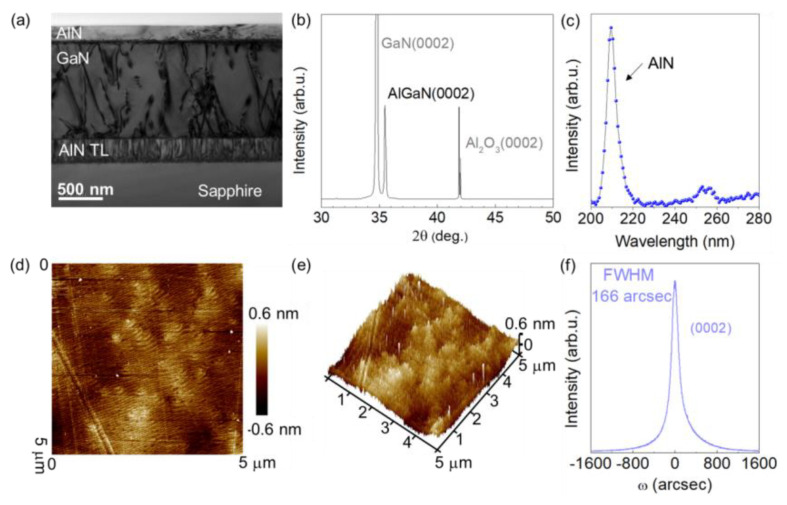
(**a**) Brightfield cross-sectional TEM images of AlN/GaN heterojunctions grown by MOCVD. (**b**) XRD pattern of the top AlN/GaN heterojunctions. (**c**) PL spectrum of the top AlN epitaxial layer. (**d**) AFM image of the AlN surface morphology. (**e**) Three-dimensional AFM surface topography of the AlN epitaxial layer. (**f**) ω-scan rocking curve of the AlN (0002) symmetric plane.

**Figure 2 nanomaterials-12-04169-f002:**
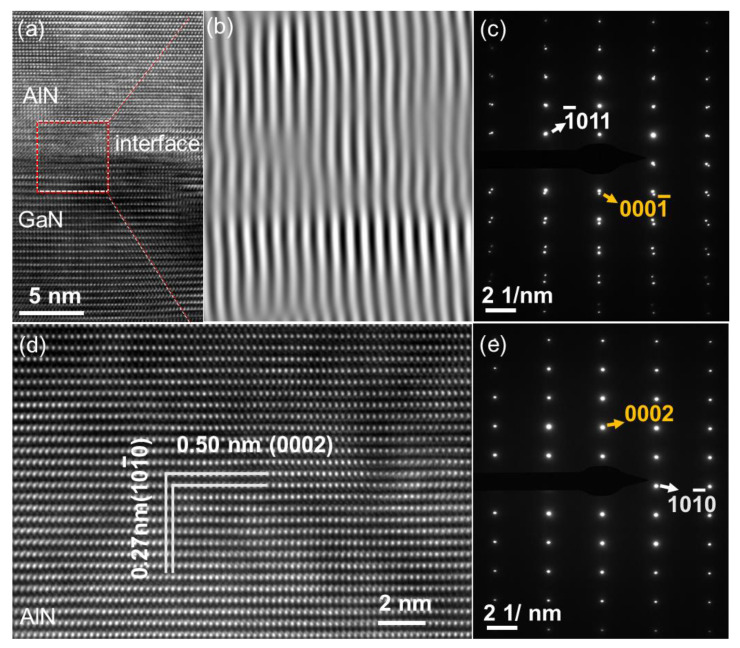
(**a**) Cross-sectional HRTEM image of the AlN/GaN interface. (**b**) Filtered images from masked fast Fourier transformation of the selected area in (**a**), showing misfit dislocations of (10$\bar{1}$0) planes. (**c**) SAED diagram at the interface. (**d**) HRTEM and (**e**) SAED diagram of the AlN photosensitive layer.

**Figure 3 nanomaterials-12-04169-f003:**
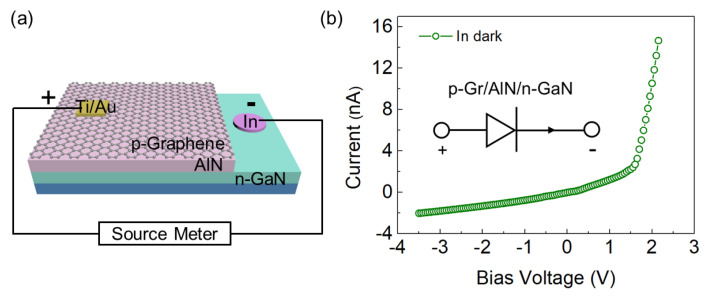
(**a**) Schematic diagram of a photovoltaic detector with a “sandwich” like structure of a p-graphene/i-AlN/n-GaN heterojunction. (**b**) I-V characteristic curves of p-graphene/i-AlN/n-GaN heterostructures under dark conditions. Illustration is the equivalent circuit of the device.

**Figure 4 nanomaterials-12-04169-f004:**
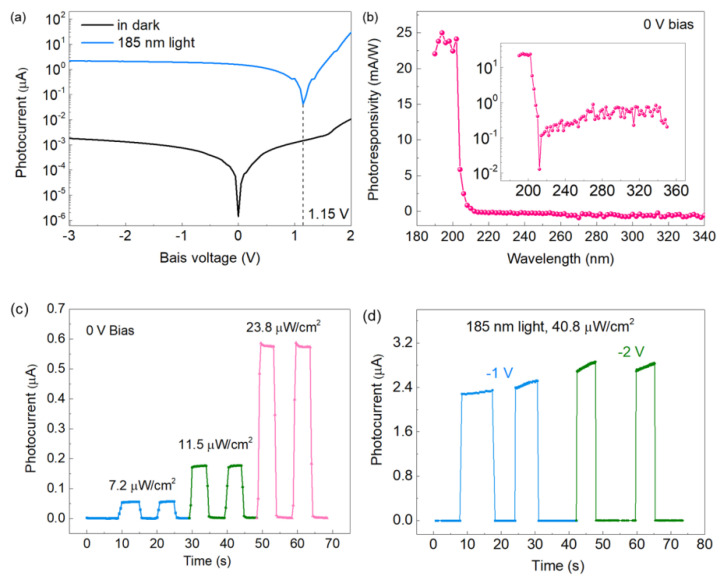
(**a**) The I-V characteristic curve of the FUV PDs under dark conditions and 185 nm light of 30.6 μW/cm^2^. The current value is treated as an absolute value. (**b**) Spectral responsivity (R_λ_) curves of devices illuminated by monochromatic light with different wavelengths. The inset is the R_λ_-λ curve in logarithmic coordinates. (**c**) Photocurrent switching characteristic curve of the device under varying power light excitation at 0 V bias. (**d**) The I_ph_-*t* curves of the device under varying bias voltage with the same incident light power of 40.8 μW/cm^2^.

**Table 1 nanomaterials-12-04169-t001:** Comparisons of UWBSs-based FUV PDs.

Material	Structure	Responsivity (A/W)	EQE (%)	Reference
Diamond film	MSM	0.027 (200 nm, 5 V)	17	[32]
c-BN film	MSM	0.032 (180 nm, 35 V)	22	[33]
Few-layered h-BN	MSM	0.0001 (212 nm, 20 V)	0.06	[34]
AlN film	MSM	0.0045 (170 nm, 30 V) 0.003 (115 nm, 30 V) 0.0085 (75 nm, 30 V)	3 3 14	[5]
MgGaO film	PIN	0.015 (185 nm, 5 V)	10	[35]
	PIN	0.01 (185 nm, 0 V)	7	[36]
AlN film	PIN	0.025 (194 nm, 0 V)	16	This work

## Data Availability

The data presented in this study are available on request from the corresponding author.

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
