# Peer review of "Heteroepitaxy Growth and Characterization of High-Quality AlN Films for Far-Ultraviolet Photodetection"

_nanomaterials, 2022, doi:10.3390/nano12234169_

Round 1

Reviewer 1 Report

The author fabricated crack-free ALN films on n-type GaN and combined it with p-type Graphene to form a PIN structure far-ultraviolet photodetector. 
XRD, AFM and TEM characterization showed the high quality of the films. The photodetector device showed responses to far-ultraviolet light with good sensitivity. While I have few doubts to be answered. 
1.  Why Sapphire is used as the base substrate for this work?
    the idea of this work is to create the innovative MOCVD method to get       ALN thin films. The key here as far as I understand is to add the ALN buffer layer between GaN and sapphire. But what if the epitaxial growth of AlN on top of GaN substrates directly? Hope the author can add more details about the novelty and importance of the 'two-step' method from work. 
2. What is the size of Ti/Au electrodes?
 Although it has been indicated that graphene has the size of 5mm square, but as a vertical photodetector device, the size should be determined by the electrodes' size. And I would suggest the author to use mA/cm2 as the unit in the photodetector electrical characterization. 
3. Line 136, the statement of Voc is apparently not true. 
 Efficiency is determined by Voc, Jsc, and fill factor. The language is suggested to be polished. 

Author Response

Reviewer #1

The author fabricated crack-free AlN films on n-type GaN and combined it with p-type Graphene to form a PIN structure far-ultraviolet photodetector.
XRD, AFM and TEM characterization showed the high quality of the films. The photodetector device showed responses to far-ultraviolet light with good sensitivity. While I have few doubts to be answered.

We would like to thank you for reviewing our paper. We have revised the manuscript according to your suggestions and believe that the revisions have improved the paper.

Please find below our responses (in blue) to each specific comment (in black) provided by the reviewer.

  1. Why Sapphire is used as the base substrate for this work?
    The idea of this work is to create the innovative MOCVD method to get ALN thin films. The key here as far as I understand is to add the ALN buffer layer between GaN and sapphire. But what if the epitaxial growth of AlN on top of GaN substrates directly? Hope the author can add more details about the novelty and importance of the 'two-step' method from work.

Reply: Thank you for the question. Highly conductive GaN single crystal substrates are expensive and not easily available, while commercial sapphire substrates are cheap. And GaN template on sapphire are often used as conductive substrates for fabricating some optoelectronic devices.

We have added more details about the novelty and importance of the 'two-step' method from work in the revised manuscript (line 7-14, paragraph 2, page 4).

  1. What is the size of Ti/Au electrodes?
    Although it has been indicated that graphene has the size of 5mm square, but as a vertical photodetector device, the size should be determined by the electrodes' size. And I would suggest the author to use mA/cm2 as the unit in the photodetector electrical characterization.

Reply: Thank you for the question. In this device structure, graphene acts as a top electrode, and its area determines the effective area of the device. While the role of Ti/Au electrodes is only collecting the top electrode carriers. They are circular electrodes with 300 μm diameter and uniformly distributed on graphene.

We have added the size of Ti/Au electrodes in the experimental section of revised manuscript.

  1. Line 136, the statement of Voc is apparently not true.
    Efficiency is determined by Voc, Jsc, and fill factor. The language is suggested to be polished.

Reply: We thank the reviewer for raising the point. We have modified the description of the Voc in the revised manuscript (line 4-5, paragraph 2, page 6).

Reviewer 2 Report

This article on "Heteroepitaxy growth and characterization of high-quality AlN films for far ultraviolet photodetection" is quite well-written and clear.
The characterization part is improvable.
1/ Why no SEM? no plan view TEM ? to be done in the future even if not possible here as the sample has been processed
2/ The deduced dislocation density is quite small. Why do not use an approach corresponding to this kind of material?
see for example :
Booker, I., Rahimzadeh Khoshroo, L., Woitok, J. F., Kaganer, V., Mauder, C., Behmenburg, H., ... & Jansen, R. H. (2010). Dislocation density assessment via X‐ray GaN rocking curve scans. physica status solidi c, 7(7‐8), 1787-1789.
Heying, B., Wu, X. H., Keller, S., Li, Y., Kapolnek, D., Keller, B. P., ... & Speck, J. S. (1996). Role of threading dislocation structure on the x‐ray diffraction peak widths in epitaxial GaN films. Applied physics letters, 68(5), 643-645.
The shortcut to determine the dislocation density should be explained. 6.0x10^7 cm-2 is quite small.
3/ figure 1, add a clear label for the ordinate axis (X-Ray intensity (arb. u), PL intensity, ...)
Usually, arbitrary units are arb. u. and not a.u. (reserve in the international unit to something else, i.e., astronomical unit)
4/ In the last part, in figure 4a, the ordinate axis could be in µA with the surface flux used on the figure to help the comparison with figure 4c.
5/ Table 1: what is the meaning of 0V? What is important for me is the reverse bias value.
After corrections, this article will become suitable to be published in nanomaterials.

Author Response

Reviewer #2

This article on "Heteroepitaxy growth and characterization of high-quality AlN films for far ultraviolet photodetection" is quite well-written and clear.
The characterization part is improvable.

We would like to thank you for reviewing our paper. We have revised the manuscript according to your suggestions and believe that the revisions have improved the paper.

Please find below our responses (in blue) to each specific comment (in black) provided by the reviewer.

1/ Why no SEM? no plan view TEM ? to be done in the future even if not possible here as the sample has been processed

Reply: Thank you for the question. SEM could not clearly identify the detailed information of AlN buffer and AlN transition layer (Figure 1a). While the cross-sectional TEM can not only characterize the layer-by-layer structure, but also more clearly characterize the difference of AlN buffer and AlN transition layer, as shown in Figure 1b.

We will conduct plan view TEM in our future work for in-depth research.

Figure 1. Cross-sectional (a) SEM (b) Bright field TEM images of AlN/GaN heterojunctions.

2/ The deduced dislocation density is quite small. Why do not use an approach corresponding to this kind of material?
see for example :
Booker, I., Rahimzadeh Khoshroo, L., Woitok, J. F., Kaganer, V., Mauder, C., Behmenburg, H., ... & Jansen, R. H. (2010). Dislocation density assessment via X‐ray GaN rocking curve scans. physica status solidi c, 7(7‐8), 1787-1789.
Heying, B., Wu, X. H., Keller, S., Li, Y., Kapolnek, D., Keller, B. P., ... & Speck, J. S. (1996). Role of threading dislocation structure on the x‐ray diffraction peak widths in epitaxial GaN films. Applied physics letters, 68(5), 643-645.
The shortcut to determine the dislocation density should be explained. 6.0x10^7 cm-2 is quite small.

Reply: Thank you for the question. We have added more detailed calculation process of dislocation density according to the literature listed by the reviewer in the revised manuscript (line 8-10, paragraph 1, page 5).

3/ figure 1, add a clear label for the ordinate axis (X-Ray intensity (arb. u), PL intensity, ...)
Usually, arbitrary units are arb. u. and not a.u. (reserve in the international unit to something else, i.e., astronomical unit)

Reply: Thank you for your suggestion. We have done this in the revised manuscript.

4/ In the last part, in figure 4a, the ordinate axis could be in µA with the surface flux used on the figure to help the comparison with figure 4c.
Reply: Thanks. We have done this in the revised manuscript. 

5/ Table 1: what is the meaning of 0V? What is important for me is the reverse bias value.

After corrections, this article will become suitable to be published in nanomaterials.

Reply: “0 V” is designed to illustrate the photovoltaic (self-powered, without bias) performance of the device. The reverse bias will increase the device performance.

Reviewer 3 Report

Review for paper «Heteroepitaxy growth and characterization of high-quality AlN films for far ultraviolet photodetection»

The article reports on obtaining high-quality AlN layers grown on GaN/AlN/Sapphire layers. Growth was carried out by the MOCVD method. Structural studies carried out using transmission electron microscopy, X-ray diffraction, atomic force microscopy, and photoluminescence spectroscopy have shown a rather low density of threading dislocations (compared to the world's best samples), as well as a high level of structural perfection of the resulting AlN layers. On the basis of the obtained heterostructures, a prototype UV photodiode was created, which is not inferior, but in some ways even superior to the known photodiodes.

I believe that the article has a high applied value and should be published. However, in terms of formulating scientific novelty and presenting structural studies, there are some remarks:

  1. As follows from the text of the Introduction, the novelty of the approach to AlN/GaN growth proposed by the authors consists of isothermal AlN/GaN growth. Since such a solution is extremely obvious and one can say “lies on the surface”, it would be appropriate to discuss why it was not previously used to grow high-quality AlN layers. Perhaps the approach is fraught with some hidden problems that are significant in other areas of application of AlN layers, but are not important for creating photodiodes?
  2. The presentation of the experimental results is somewhat confusing. To improve the perception of the material, I would recommend separating the presentation of the results and their discussion. This will allow the reader to easily separate the objective experimental data from the author's interpretations.

In addition, there are a number of minor remarks in the text:

  1. String 28. The explain the vertical and horizontal FD scheme is needed. It is necessary for wide range readers.
  2. String 35. The modality “only 2.4%” is not very appropriate. Because, that 2, that 10% lattice mismatch does not leave a chance to obtain pseudomorphic layers of sufficient thickness (at least 100 nm). The experience of growing various III-V compounds on silicon shows that the final dislocation density is determined by many factors, and not only by the value of the mismatch of the lattice constants.
  3. String 38. A remark has a purely grammatical nature. It would be appropriate to say that the mismatch affects the crystal quality not only of AlN, but also of GaN.
  4. Moving the experimental section to the end of the article does not seem justified. This only complicates reading and generates unnecessary questions for the reader. So, for example, at the first reading of the article, it is clear that the description of the TEM data begins on string 70. But it is not clear what kind of structure we are talking about - about the structure grown in the framework of this work, or whether the discussion of literature data (which was discussed above) continues.
  5. String 76. It is not indicated under what conditions the PL spectra were measured (temperature, power density). Alas, even reading the experimental section did not clarify the situation.
  6. 6A conclusion was made about the high crystalline quality of the AlN layers without comparing the PL spectra of the studied sample and any reference sample, which seems unconvincing. It is recommended to pay attention to the results of [Appl. Phys. Lett. 102, 211107 (2013); https://doi.org/10.1063/1.4807485], which describes a method for determining the internal quantum yield of luminescence in heterostructures. The technique is based on measuring the intensity dependence of the PL spectra.
  7. Do I understand correctly that the 3D surface topography obtained by AFM is simply a 3D representation of the same area of ​​the surface as shown in the 2D color-graded image? If so, Figure 1e seems redundant. The absence of cracks is also clearly visible in Figure 1d.
  8. In Figure 1b, there is apparently a typo, the reflections are signed as (002), instead of (0002).
  9. Do I understand correctly that the absence of periodicity violations in Figure 2b shows that AlN has grown pseudomorphically stressed? At least in the limit of image 2b. Other data in Fig. 2 also speak about this. Taking into account the density of threading dislocations in AlN of about 6x107 cm-2, does this mean that dislocations could begin to appear in layers grown later than those shown in Fig. 2?
  10. For a more understandable description of the obtained PDs, it would be good to show their band diagram. This would make it possible to visualize the operation of the device under discussion and understand all its advantages.
  11. The nature of the spectral selectivity of PD is not clear from the text of the article. From the side of long waves, absorption is limited by the band gap of AlN. And what leads to the limitation of absorption by short wavelengths?
  12.  String 190. Nothing is said about the transverse dimensions of the Ti and Au layers. Do they match the dimensions of the graphene sheet? Or is it much less, and then the light mainly falls on the graphene contact?

Author Response

Reviewer #3

The article reports on obtaining high-quality AlN layers grown on GaN/AlN/Sapphire layers. Growth was carried out by the MOCVD method. Structural studies carried out using transmission electron microscopy, X-ray diffraction, atomic force microscopy, and photoluminescence spectroscopy have shown a rather low density of threading dislocations (compared to the world's best samples), as well as a high level of structural perfection of the resulting AlN layers. On the basis of the obtained heterostructures, a prototype UV photodiode was created, which is not inferior, but in some ways even superior to the known photodiodes.

I believe that the article has a high applied value and should be published. However, in terms of formulating scientific novelty and presenting structural studies, there are some remarks:

We would like to thank you for reviewing our paper. We have revised the manuscript according to your suggestions and believe that the revisions have improved the paper.

Please find below our responses (in blue) to each specific comment (in black) provided by the reviewer.

  1. As follows from the text of the Introduction, the novelty of the approach to AlN/GaN growth proposed by the authors consists of isothermal AlN/GaN growth. Since such a solution is extremely obvious and one can say “lies on the surface”, it would be appropriate to discuss why it was not previously used to grow high-quality AlN layers. Perhaps the approach is fraught with some hidden problems that are significant in other areas of application of AlN layers, but are not important for creating photodiodes?

Reply: Thanks for the reviewer's suggestions. The novelty of the approach of this work is indeed isothermal AlN/GaN growth, and such a solution lies on the surface. The previous research rarely covers the application requirements for epitaxial AlN on GaN, which is different from the current requirement of this work: the photovoltaic detector with special AlN/GaN vertical device structure is constructed by comprehensively considering many factors of substrate selection. If the epitaxial parameters of AlN and GaN are not well controlled respectively, it is difficult to realize the transition of growth mode from 3D to 2D, let alone the atomically flat surface. It requires a lot of exploration of epitaxial conditions.

This isothermal approach is also very important for creating photodiodes: too high a temperature will lead to the decomposition of GaN, while too low a temperature will not allow AlN atoms to obtain enough energy for lateral diffusion, which leads to the failure of growth mode transition. Thus, it will affect the photo-response performance of the photodiodes, especially the leakage current or dark current of the device.

According to the reviewer's suggestion, we have added some appropriate background discussion on “why it was not previously used to grow high-quality AlN layers” in the revised manuscript (line 5-7, paragraph 2, page 4).

  1. The presentation of the experimental results is somewhat confusing. To improve the perception of the material, I would recommend separating the presentation of the results and their discussion. This will allow the reader to easily separate the objective experimental data from the author's interpretations.

Reply: Thanks for your suggestions. Presenting the results separately from the discussion may be appropriate for articles with more centralized data structures. For this paper, there are both material characterization and device performance test. If the results are given first and then discussed, it will inevitably lead to repeated description. In addition, putting different results together can make readers feel disconnected. Therefore, we think it is more appropriate to describe the discussion together with the results for the data structure of this paper.

In addition, there are a number of minor remarks in the text:

  1. String 28. The explain the vertical and horizontal FD scheme is needed. It is necessary for wide range readers.

Reply: Thank you for your suggestion. The difference between the two structures lies in whether the carrier transmission channel between the two electrodes in the prepared device structure is in the horizontal direction or the vertical direction, that is, it depends on the carrier transmission direction. For example, the typical cross-fingered electrode deposited on a semiconductor is horizontal structure.

A brief description has been added to the revised manuscript (line 6-7, paragraph 1, page 2).

  1. String 35. The modality “only 2.4%” is not very appropriate. Because, that 2, that 10% lattice mismatch does not leave a chance to obtain pseudomorphic layers of sufficient thickness (at least 100 nm). The experience of growing various III-V compounds on silicon shows that the final dislocation density is determined by many factors, and not only by the value of the mismatch of the lattice constants.

Reply: Thanks. We have revised the relevant description in the revised manuscript (line 13-14, paragraph 1, page 2).

.

  1. String 38. A remark has a purely grammatical nature. It would be appropriate to say that the mismatch affects the crystal quality not only of AlN, but also of GaN.

Reply: Thanks. We have revised the relevant description in the revised manuscript (line 3, paragraph 2, page 2).

  1. Moving the experimental section to the end of the article does not seem justified. This only complicates reading and generates unnecessary questions for the reader. So, for example, at the first reading of the article, it is clear that the description of the TEM data begins on string 70. But it is not clear what kind of structure we are talking about - about the structure grown in the framework of this work, or whether the discussion of literature data (which was discussed above) continues.

Reply: Thanks for raising the point. We've moved the experiment section to appropriate place of the main text (page 3-4).

  1. String 76. It is not indicated under what conditions the PL spectra were measured (temperature, power density). Alas, even reading the experimental section did not clarify the situation.

Reply: Thank you for raising the point. PL spectra are measured at room temperature with a laser power of 0.5 W@50Hz. Relevant descriptions have been added to the experimental section.

  1. 6A conclusion was made about the high crystalline quality of the AlN layers without comparing the PL spectra of the studied sample and any reference sample, which seems unconvincing. It is recommended to pay attention to the results of [Appl. Phys. Lett. 102, 211107 (2013); https://doi.org/10.1063/1.4807485], which describes a method for determining the internal quantum yield of luminescence in heterostructures. The technique is based on measuring the intensity dependence of the PL spectra.

Reply: Thanks for raising the point. The PL spectra provided in this work is the band-edge emission of AlN single crystalline film, which is different from the widely reported AlN defect emission. And the very narrow FWHM of 5.2 nm is comparable to currently reported state-of-the-art result (ACS Nano 2018,12, 425), suggesting its high crystalline quality.

According to your suggestion, we have added an appropriate comparison in the revised manuscript (line 23-27, paragraph 2, page 4).

  1. Do I understand correctly that the 3D surface topography obtained by AFM is simply a 3D representation of the same area of the surface as shown in the 2D color-graded image? If so, Figure 1e seems redundant. The absence of cracks is also clearly visible in Figure 1d.

Reply: Thanks for raising the point. We think the 3D surface topography will not be redundant due to the fact that 3D morphology image can more intuitively present surface information such as the state of ups and downs, depth variation, roughness and granularity than that of 2D.

  1. In Figure 1b, there is apparently a typo, the reflections are signed as (002), instead of (0002).

Reply: Thanks. This typo has been modified. 

  1. Do I understand correctly that the absence of periodicity violations in Figure 2b shows that AlN has grown pseudomorphically stressed? At least in the limit of image 2b. Other data in Fig. 2 also speak about this. Taking into account the density of threading dislocations in AlN of about 6x107cm-2, does this mean that dislocations could begin to appear in layers grown later than those shown in Fig. 2?

Reply: Exactly correct. The stress has not been fully released in the framed area, and the dislocation could begin to appear in layers grown later.

  1. For a more understandable description of the obtained PDs, it would be good to show their band diagram. This would make it possible to visualize the operation of the device under discussion and understand all its advantages.

Reply: For the present complex device structure, an accurate band structure requires a number of accurate parameters, which is quite difficult. If we only draw a brief diagram (non-quantitative, as shown below), it may not contribute to the readability of this article.

  1. The nature of the spectral selectivity of PD is not clear from the text of the article. From the side of long waves, absorption is limited by the band gap of AlN. And what leads to the limitation of absorption by short wavelengths?

Reply: Light with wavelengths below 190 nm is strongly absorbed by the air, making it difficult to measure the spectral response in the short wavelength region. We believe the device still has a strong response to light below 190 nm. However, the response data of shorter wavelength cannot be obtained due to the limitation of our test conditions.

The absorption of short wavelength is mainly limited by the penetration depth of the band, which is the main reason for the cut-off of spectral response at short wavelength.

  1. String 190. Nothing is said about the transverse dimensions of the Ti and Au layers. Do they match the dimensions of the graphene sheet? Or is it much less, and then the light mainly falls on the graphene contact?

Reply: Thank you for the question. In this device structure, graphene acts as a top electrode, and its area determines the effective area of the device. While the Ti/Au electrodes are only used to collect the top electrode carriers, which are circular 300 μm diameter electrodes uniformly distributed on graphene.

We have added the size of Ti/Au electrodes in the experimental section of revised manuscript.

Round 2

Reviewer 3 Report

Reviewer #3

The article reports on obtaining high-quality AlN layers grown on GaN/AlN/Sapphire layers. Growth was carried out by the MOCVD method. Structural studies carried out using transmission electron microscopy, X-ray diffraction, atomic force microscopy, and photoluminescence spectroscopy have shown a rather low density of threading dislocations (compared to the world's best samples), as well as a high level of structural perfection of the resulting AlN layers. On the basis of the obtained heterostructures, a prototype UV photodiode was created, which is not inferior, but in some ways even superior to the known photodiodes.

I believe that the article has a high applied value and should be published. However, in terms of formulating scientific novelty and presenting structural studies, there are some remarks:

We would like to thank you for reviewing our paper. We have revised the manuscript according to your suggestions and believe that the revisions have improved the paper.

Please find below our responses (in blue) to each specific comment (in black) provided by the reviewer.

  1. As follows from the text of the Introduction, the novelty of the approach to AlN/GaN growth proposed by the authors consists of isothermal AlN/GaN growth. Since such a solution is extremely obvious and one can say “lies on the surface”, it would be appropriate to discuss why it was not previously used to grow high-quality AlN layers. Perhaps the approach is fraught with some hidden problems that are significant in other areas of application of AlN layers, but are not important for creating photodiodes?

Reply: Thanks for the reviewer's suggestions. The novelty of the approach of this work is indeed isothermal AlN/GaN growth, and such a solution lies on the surface. The previous research rarely covers the application requirements for epitaxial AlN on GaN, which is different from the current requirement of this work: the photovoltaic detector with special AlN/GaN vertical device structure is constructed by comprehensively considering many factors of substrate selection. If the epitaxial parameters of AlN and GaN are not well controlled respectively, it is difficult to realize the transition of growth mode from 3D to 2D, let alone the atomically flat surface. It requires a lot of exploration of epitaxial conditions.

This isothermal approach is also very important for creating photodiodes: too high a temperature will lead to the decomposition of GaN, while too low a temperature will not allow AlN atoms to obtain enough energy for lateral diffusion, which leads to the failure of growth mode transition. Thus, it will affect the photo-response performance of the photodiodes, especially the leakage current or dark current of the device.

According to the reviewer's suggestion, we have added some appropriate background discussion on “why it was not previously used to grow high-quality AlN layers” in the revised manuscript (line 5-7, paragraph 2, page 4).

I recommend discussing this topic in the Introduction section. Because in the Introduction it is necessary to show the investigation motivation and novelty of the work. Also, I recommend discussing, why the growth temperature was chosen at 1050C (GaN optimum). Why not something between 1050 and 1250 C?

  1. The presentation of the experimental results is somewhat confusing. To improve the perception of the material, I would recommend separating the presentation of the results and their discussion. This will allow the reader to easily separate the objective experimental data from the author's interpretations.

Reply: Thanks for your suggestions. Presenting the results separately from the discussion may be appropriate for articles with more centralized data structures. For this paper, there are both material characterization and device performance test. If the results are given first and then discussed, it will inevitably lead to repeated description. In addition, putting different results together can make readers feel disconnected. Therefore, we think it is more appropriate to describe the discussion together with the results for the data structure of this paper.

I meant the experimental data concerning the growth of AlN layers. Of course, the section on the photodiode stands apart and is an adornment of the work in applied terms. I still recommend revising the structure of the presentation of the results (Figures 1 and 2) in favor of a separate presentation of the results and their discussion. Again, this is only my recommendation, the decision on the final form of the article remains with the authors.

In addition, there are a number of minor remarks in the text:

  1. String 28. The explain the vertical and horizontal FD scheme is needed. It is necessary for wide range readers.

Reply: Thank you for your suggestion. The difference between the two structures lies in whether the carrier transmission channel between the two electrodes in the prepared device structure is in the horizontal direction or the vertical direction, that is, it depends on the carrier transmission direction. For example, the typical cross-fingered electrode deposited on a semiconductor is horizontal structure.

A brief description has been added to the revised manuscript (line 6-7, paragraph 1, page 2).

Ok

  1. String 35. The modality “only 2.4%” is not very appropriate. Because, that 2, that 10% lattice mismatch does not leave a chance to obtain pseudomorphic layers of sufficient thickness (at least 100 nm). The experience of growing various III-V compounds on silicon shows that the final dislocation density is determined by many factors, and not only by the value of the mismatch of the lattice constants.

Reply: Thanks. We have revised the relevant description in the revised manuscript (line 13-14, paragraph 1, page 2).

Ok

  1. String 38. A remark has a purely grammatical nature. It would be appropriate to say that the mismatch affects the crystal quality not only of AlN, but also of GaN.

Reply: Thanks. We have revised the relevant description in the revised manuscript (line 3, paragraph 2, page 2).

Ok

  1. Moving the experimental section to the end of the article does not seem justified. This only complicates reading and generates unnecessary questions for the reader. So, for example, at the first reading of the article, it is clear that the description of the TEM data begins on string 70. But it is not clear what kind of structure we are talking about - about the structure grown in the framework of this work, or whether the discussion of literature data (which was discussed above) continues.

Reply: Thanks for raising the point. We've moved the experiment section to appropriate place of the main text (page 3-4).

Ok. But it strange for me to move the Fig. 4 to this section too. At the very least, the numbering of the figures needs to be corrected.

  1. String 76. It is not indicated under what conditions the PL spectra were measured (temperature, power density). Alas, even reading the experimental section did not clarify the situation.

Reply: Thank you for raising the point. PL spectra are measured at room temperature with a laser power of 0.5 W@50Hz. Relevant descriptions have been added to the experimental section.

For PL measurements, it is not the total radiation power that is important, but the power density (W/cm2). It is recommended to specify this value.

Also, I would specify laser wavelength.

  1. A conclusion was made about the high crystalline quality of the AlN layers without comparing the PL spectra of the studied sample and any reference sample, which seems unconvincing. It is recommended to pay attention to the results of [Appl. Phys. Lett. 102, 211107 (2013); https://doi.org/10.1063/1.4807485], which describes a method for determining the internal quantum yield of luminescence in heterostructures. The technique is based on measuring the intensity dependence of the PL spectra.

Reply: Thanks for raising the point. The PL spectra provided in this work is the band-edge emission of AlN single crystalline film, which is different from the widely reported AlN defect emission. And the very narrow FWHM of 5.2 nm is comparable to currently reported state-of-the-art result (ACS Nano 2018,12, 425), suggesting its high crystalline quality.

According to your suggestion, we have added an appropriate comparison in the revised manuscript (line 23-27, paragraph 2, page 4).

I cannot agree that only the width of the interband PL band indicates the crystalline quality of the material. First of all, the quality of the material is indicated by the intensity of radiative recombination. Many defects (both extended and point defects) can form centers of nonradiative recombination, which will adversely affect the PL intensity. The width of the PL band undoubtedly reflects the quality of the material, but only in terms of the level of disorder (which is more typical for solid solutions of the AlGaN type) due to the formation of tails of the density of states.

Since PL measurements are always relative, it is extremely important to compare the test sample with the test sample. If it is not possible perform such measurements, I would recommend softening the conclusion from the PL data and restricting ourselves to the fact that the rate of nonradiative recombination in the structures is not so high as to completely suppress radiative recombination.

  1. Do I understand correctly that the 3D surface topography obtained by AFM is simply a 3D representation of the same area of the surface as shown in the 2D color-graded image? If so, Figure 1e seems redundant. The absence of cracks is also clearly visible in Figure 1d.

Reply: Thanks for raising the point. We think the 3D surface topography will not be redundant due to the fact that 3D morphology image can more intuitively present surface information such as the state of ups and downs, depth variation, roughness and granularity than that of 2D.

Ok

  1. In Figure 1b, there is apparently a typo, the reflections are signed as (002), instead of (0002).

Reply: Thanks. This typo has been modified.

  1. Do I understand correctly that the absence of periodicity violations in Figure 2b shows that AlN has grown pseudomorphically stressed? At least in the limit of image 2b. Other data in Fig. 2 also speak about this. Taking into account the density of threading dislocations in AlN of about 6x107 cm-2, does this mean that dislocations could begin to appear in layers grown later than those shown in Fig. 2?

Reply: Exactly correct. The stress has not been fully released in the framed area, and the dislocation could begin to appear in layers grown later.

Ok

  1. For a more understandable description of the obtained PDs, it would be good to show their band diagram. This would make it possible to visualize the operation of the device under discussion and understand all its advantages.

Reply: For the present complex device structure, an accurate band structure requires a number of accurate parameters, which is quite difficult. If we only draw a brief diagram (non-quantitative, as shown below), it may not contribute to the readability of this article.

Ok

  1. The nature of the spectral selectivity of PD is not clear from the text of the article. From the side of long waves, absorption is limited by the band gap of AlN. And what leads to the limitation of absorption by short wavelengths?

Reply: Light with wavelengths below 190 nm is strongly absorbed by the air, making it difficult to measure the spectral response in the short wavelength region. We believe the device still has a strong response to light below 190 nm. However, the response data of shorter wavelength cannot be obtained due to the limitation of our test conditions.

The absorption of short wavelength is mainly limited by the penetration depth of the band, which is the main reason for the cut-off of spectral response at short wavelength.

Ok

  1. String 190. Nothing is said about the transverse dimensions of the Ti and Au layers. Do they match the dimensions of the graphene sheet? Or is it much less, and then the light mainly falls on the graphene contact?

Reply: Thank you for the question. In this device structure, graphene acts as a top electrode, and its area determines the effective area of the device. While the Ti/Au electrodes are only used to collect the top electrode carriers, which are circular 300 μm diameter electrodes uniformly distributed on graphene.

We have added the size of Ti/Au electrodes in the experimental section of revised manuscript.

Ok

Some novel comments.

In the text of the paper, the authors refer to [Wei Zheng, Richeng Lin, Junxue Ran, Zhaojun Zhang, Xu Ji, and Feng Huang ACS Nano 2018, 12, 425 https://doi.org/10.1021/acsnano.7b06633] in the context of that this is their previous work. However, the list of authors of these two works does not overlap at all, as well as the list of affiliations. To my deep regret, during the initial work with the text of the authors' paper, I did not pay attention to this paper. However, now, with a detailed acquaintance with it, I clearly see that the work of the authors largely repeats this paper.

I strongly recommend authors to radically reconsider their vision of the work. I believe that the work should be restructured precisely in the key of data reproduction [ACS Nano 2018, 12, 425]. At the same time, a convincing justification for the relevance of such reproduction should be given. For example, this may be the instability of the characteristics of photodiodes discussed in [ACS Nano 2018, 12, 425] and their poor reproducibility.
